# Identification of Resveratrol as Bioactive Compound of Propolis from Western Romania and Characterization of Phenolic Profile and Antioxidant Activity of Ethanolic Extracts

**DOI:** 10.3390/molecules24183368

**Published:** 2019-09-16

**Authors:** Alexandra Duca, Adrian Sturza, Elena-Alina Moacă, Monica Negrea, Virgil-Dacian Lalescu, Diana Lungeanu, Cristina-Adriana Dehelean, Danina-Mirela Muntean, Ersilia Alexa

**Affiliations:** 1Doctoral School Medicine-Pharmacy, “Victor Babeş” University of Medicine and Pharmacy of Timişoara, Eftimie Murgu Sq. No. 2, 300041 Timişoara, Romania; alexandra_duca@yahoo.com; 2Faculty of Medicine, “Victor Babeş” University of Medicine and Pharmacy of Timişoara, Eftimie Murgu Sq. No. 2, 300041 Timişoara, Romania; sturza.adrian@umft.ro (A.S.); dlungeanu@umft.ro (D.L.); 3Center for Translational Research & Systems Medicine, “Victor Babeş” University of Medicine and Pharmacy of Timişoara, Eftimie Murgu Sq. No. 2, 300041 Timişoara, Romania; 4Faculty of Pharmacy, “Victor Babeş”, University of Medicine and Pharmacy of Timişoara, Eftimie Murgu Sq., No. 2, 300041 Timişoara, Romania; alina.moaca@umft.ro; 5Faculty of Food Engineering, Banat University of Agricultural Sciences and Veterinary Medicine “King Michael I of Romania”of Timişoara, Calea Aradului, No. 119, 300645 Timișoara, Romania; negrea_monica2000@yahoo.com (M.N.); dlalescu@yahoo.com (V.-D.L.); alexa.ersilia@yahoo.ro (E.A.)

**Keywords:** Western Romanian propolis, polyphenols, resveratrol, Folin-Ciocȃlteu, LC-MS, DPPH method, FOX assay

## Abstract

The present study aimed to assess the phenolic content of eight ethanolic propolis samples (P1–P8) harvested from different regions of Western Romania and their antioxidant activity. The mean value of total phenolic content was 214 ± 48 mg gallic acid equivalents (GAE)/g propolis. All extracts contained kaempferol (514.02 ± 114.80 μg/mL), quercetin (124.64 ± 95.86 μg/mL), rosmarinic acid (58.03 ± 20.08 μg/mL), and resveratrol (48.59 ± 59.52 μg/mL) assessed by LC-MS. The antioxidant activity was evaluated using 2 methods: (i) DPPH (2,2-diphenyl-1-picrylhydrazyl) assay using ascorbic acid as standard antioxidant and (ii) FOX (Ferrous iron xylenol orange OXidation) assay using catalase as hydrogen peroxide (H_2_O_2_) scavenger. The DPPH radical scavenging activity was determined for all samples applied in 6 concentrations (10, 5, 3, 1.5, 0.5 and 0.3 mg/mL). IC_50_ varied from 0.0700 to 0.9320 mg/mL (IC_50_ of ascorbic acid = 0.0757 mg/mL). The % of H_2_O_2_ inhibition in FOX assay was assessed for P1, P2, P3, P4 and P8 applied in 2 concentrations (5 and 0.5 mg/mL). A significant H_2_O_2_% inhibition was obtained for these samples for the lowest concentration. We firstly report the presence of resveratrol as bioactive compound in Western Romanian propolis. The principal component analysis revealed clustering of the propolis samples according to the polyphenolic profile similarity.

## 1. Introduction

Propolis is a sticky material harvested and processed by honey bees from buds, leaves, and the bark of trees (e.g., poplar, cypress, pine, birch, alder, etc.) in order to secure and defend the hives (from Gr., *pro*—in front/in defense of, *polis*—city/hive). This natural resinous mixture has been extensively used by ancient civilizations as folk medicine for its numerous beneficial properties that are synergistically related to its complex chemical composition [1,2].Generally, the poplar type of propolis that predominates in temperate zones (originating mainly from the buds of *Populus nigra* L.), comprises 50% resins, 30% wax, 10% volatile oils, 5% pollen, and 5% other organic constituents [2,3,4]. The bioactive compounds in propolis exert a wide range of therapeutic effects, such as: antioxidant, antibacterial, antifungal, antiviral, anti-inflammatory, anti-carcinogenic, hepato- and cardioprotective effects. These properties, in particular the antioxidant one, has been largely ascribed to the polyphenolic fraction (phenolic acids, flavonoids) of propolis [5,6].

Oxidative stress is the common pathomechanism of age-related non-communicable chronic diseases [7,8] and propolis has been widely acknowledged as a valuable source of natural antioxidants able to counteract and/or prevent it. Indeed, the antioxidant properties have been systematically reported for both ethanolic and aqueous propolis extracts [9] using traditional methods represented by the assessment of the scavenging capacity of radicals such as DPPH (2,2-diphenyl-1-picrylhydrazyl), ABTS (2,2′-azino-bis(3-ethylbenzothiazoline-6-sulphonic acid), the ferric reducing ability of plasma (FRAP), and the oxygen radical absorbance capacity (ORAC), respectively. The antioxidant capacity of propolis strongly depends on its chemical composition, which in turn is influenced by the extraction methods [10]. Phenolic acids act as antioxidants via several pathomechanisms such as free radicals scavengers, metals chelators, and inhibitors of oxidative enzymes [5]. The antioxidant capacity of propolis has been systematically investigated in vitro, in animal models and lately, also in clinical settings [9].

The present study was double aimed: (i) to assess the polyphenolic profile of eight ethanolic extracts of propolis samples harvested from different regions of Western Romania, including the presence of novel bioactive compounds, such as resveratrol, and (ii) to evaluate their antioxidant activity by 2 methods. Furthermore, in order to investigate the relationship between the phenolic profile of propolis samples and their geographical origin, principal component analysis (PCA) followed by clustering were performed.

The main finding is the identification of resveratrol as an important bioactive compound in all propolis samples collected from Western Romania. Additionally, the dose-dependent characterization by two methods of the antioxidant capacity was performed.

## 2. Results

### 2.1. Total and Individual Polyphenols

Total phenolic content (TPC) of the samples was determined spectrophotometrically (Folin-Ciocȃlteu method) and expressed as mg of gallic acid equivalents per g of sample (mg GAE/g)—Figure 1. The mean value of TPC was 214.30 ± 48.15 mg GAE/g of dry weight of propolis. The highest value of TPC was found in sample P4 (333.83 ± 13.79 mg GAE/g), whereas the lowest value was found in sample P7 (170.24 ± 0.34 mg GAE/g). The significant differences between P4 and the other samples are illustrated by asterisks.

The distribution of individual phenols in the eight samples is depicted in Table 1 and Figure 2A, respectively. Four compounds were identified in the highest (yet variable) amounts in all samples, namely, kaempferol, quercetin, resveratrol, and rosmarinic acid—as shown in Figure 2B.

Kaempferol and quercetin were the major polyphenolic compounds identified in the Romanian propolis extracts, with the former representing 72.86 ± 11.33% of all individual polyphenols, and the latter 15.97 ± 9.68, respectively. The percentage of rosmarinic acid (ester of caffeic acid) from all polyphenols identified in propolis, varied between 4.34% and 14.04%; the mean concentration in the 8 extracts was 58.03 ± 20.08 μg/mL. Resveratrol, a powerful protective stilbene derivative, was also constantly present in variable concentrations in all eight samples (Table 1) and represented between 0.63% and 19.77% of all polyphenols. Indeed, the lowest concentration was found in P6 (4.90 ± 0.57 μg/mL), whereas the highest one was present in P7 (188.50 ± 42.52 μg/mL). This is the most relevant finding of the present study and appears a characteristic of the Western Romanian propolis since it was not reported in other types of Romanian propolis. 

Chromatograms of individual samples (P1 to P8) are available as Appendix A.

### 2.2. Antioxidant Activity Assessed by DPPH (2,2-diphenyl-1-picrylhydrazyl) Method

The DPPH radical scavenging activity of the 8 propolis ethanolic extracts was determined for 6 concentrations (10 mg/mL, 5 mg/mL, 3 mg/mL, 1.5 mg/mL, 0.5 mg/mL and 0.3 mg/mL) (Table 2) and was monitored for 1200 s. In parallel, the antioxidant activity of increasing concentrations of ascorbic acid was evaluated as positive control, resulting in a 92.68% inhibition for the highest tested concentration (0.13 mg/mL). IC_50_ (the extract concentration that determines the 50% DPPH inhibition) was further calculated for each sample and expressed in mg/mL (Table 3, Figure 3).

As presented in Table 2, the maximal radical scavenging activity was obtained for the highest concentration (10 mg/mL) for almost all samples (except P6 and P7) and was similar to the effect of the standard antioxidant, ascorbic acid, applied in concentrations of −0.13 mg/mL and 0.11 mg/mL, respectively. However, similar values were also recorded for the concentration of 5 mg/mL for all the samples (except P6). This observation prompted us to further test the scavenger activity of the samples with the FOX assay using 5 mg/mL as the highest effective concentration.

The percentage of DPPH inhibition still remained high for the next two lower concentrations (3 mg/mL and 1.5 mg/mL, respectively) for all samples but P6. The antioxidant activity showed an important decrease for all samples when applied in the lowest tested concentration (0.3 mg/mL), except for P2. Indeed, sample P2 at 0.3 mg/mL still preserved a radical scavenging activity of 78.16%.

The concentration-dependency of the radical scavenging activity of the eight propolis samples is depicted in Figure 4.

The percentage of DPPH inhibition for all samples (except P6) at 10 mg/mL, 5 mg/mL and 3 mg/mL was ~90% and thus similar to the one of ascorbic acid (at 0.105–0.13 mg/mL). Moreover, all samples still retained a high antioxidant capacity (88.60% ± 1.73%) when applied at 1.5 mg/mL, except, once again, for P6 (67.58%).

Also, all samples (but P6) quickly reacted with the DPPH radical (~80 s) prior to reaching the equilibrium of reaction (that was inferior to the one of ascorbic acid) (Figure 5).

An example of the time-dependency of the reaction is presented in Figure 6 for the P2 sample that showed the highest DPPH scavenging activity among the samples; importantly, the antioxidant capacity of P2 remained high (at 78.16% of inhibition) even when tested in the lowest concentration (0.3 mg/mL).

The IC_50_ values (Table 3) varied from 0.0700 mg/mL for the P2 sample (the highest antioxidant capacity) to 0.9945 mg/mL (the lowest antioxidant capacity) in case of P6 sample. TheIC_50_ variation of the Western Romanian propolis samples is displayed in Figure 7.

The standing out value of IC_50_ for the P6 sample was an intriguing finding. Of note, this sample had the lowest amount of resveratrol (4.90 ± 0.69 μg/mL). It is tempting to speculate that the low concentration of this powerful antioxidant may partly account for the low antioxidant activity of P6. Also, P7 had a rather high value of IC_50_ (0.5039 ± 0.0234 mg/mL) that might be explained by the fact that it contained the lowest amount of total polyphenols (170.24 mg/g). However, no statistical significance was reached when we calculated the Pearson coefficient as presented in Section 2.5.

### 2.3. Antioxidant Activity Assessed by FOX (Ferrous Oxidation-Xylenol Orange) Assay

FOX assay (PeroxiDetect kit, Sigma-Aldrich) is a rapid spectrophotometrical assay based on the ability of hydroperoxides to oxidize ferrous iron (Fe^2+^) into ferric iron (Fe^3+^) which will react and form a colored complex with xylenol orange (XO) that can be measured at 560 nm. The % of H_2_O_2_ inhibition is presented for the samples that showed an important DPPH scavenging effect (P1, P2, P3, P4, P8) applied in a high (5 mg/mL) and low concentration (0.5 mg/mL), respectively. In this assay catalase (CAT), the H_2_O_2_ scavenger, was used to compare the effect of propolis samples (Figure 8).

As depicted in Figure 8, all samples inhibited the H_2_O_2_ when applied either in high or low concentrations. Interestingly, the lower concentration appeared to be more efficient in comparison to the higher one; a percentage of around 32% inhibition was obtained for all samples at 0.5 mg/mL, which represents ~50% of the CAT effect. Samples P5, P6 and P7 did not exhibit antioxidant activity assessed by the FOX technique (data not shown).

### 2.4. Principal Component Analysis (PCA) and Sample Clustering

Chemometrics represents a useful statistic tool to disclose the relation between certain constituents identified in the propolis samples and its geographic provenance [11,12]. Following the inter-sample correlation analysis (Figure 9), a PCA was conducted on the mean values of measured traits to study the parameters that mostly contributed to the total data variation. The PCA produced eight components, with the first three accounting for a total of 81.15% of the variance, as follows: 43.20%, 25.52%, and 12.48%, respectively. Based on components’ scores and loadings, the most important contributors to the first component were *p*-coumaric acid, rutin and quercetin, all being negatively correlated with this component. The second component had epicatechin as the main positive contributor, while the third component had total phenolic content as the major positive contributor. The similarity among polyphenolic profiles was examined when each sample was plotted using the first three principal components showing a clustering tendency, which allowed us to perform a cluster analysis (Figure 10).

Accordingly, propolis samples were assigned to the following representative clusters: group 1—P3 and P5 showed the highest similarity level (96.49%), group 2—P6 and P8, which showed a similarity of 96.01%, group 3—P1 and P7 with a similarity of 94.28%. Samples P1, P7, P3 and P5 presented an important level of similarity (56.10%). Samples P2 and P4 were grouped together, but the similarity between them was low (13.40%) (Figure 11).

### 2.5. Pearson Correlation Coefficients

Since the antioxidant activity of propolis extracts has been classically ascribed to the polyphenolic content, the Pearson correlation coefficient (r) was computed as a measure of the linear association between quantitative variables. No statistically significant correlations between total phenolic content of Western Romanian propolis and either technique used to assess the antioxidant capacity were found in our study. As expected, the antioxidant activity evaluated by the DPPH method showed a significant, strong negative correlation with IC_50_ (r = −0.87, *p* < 0.02); indeed, a lower IC_50_ value of propolis samples corresponds to a greater capability to neutralize free radicals. It is worth mentioning that, for the poplar propolis extracts, correlations between total polyphenols/flavonoids and antioxidant capacity were not always reported in the literature [9].

## 3. Discussion

### 3.1. Total & Individual Polyphenols and PCA Analysis

The present paper was firstly aimed at characterizing the polyphenolic composition of ethanolic propolis samples collected from different regions of Western Romania.

The TPC of Western Romanian propolis varied from the lowest value of 170.24 mg GAE/g (sample P7) to the highest one, i.e., 333.83 mg GAE/g (sample P4) with a mean value of 214.30 ± 48.15 mg GAE/g of dry weight. Our results are in line with the data reported in the international literature for the TPC of propolis extracts, with values ranging from 30 to 200 mg GAE/g [9]. However, they are different from the results of similar studies which analyzed propolis samples collected from regions in Central Romania. Indeed, in a study that investigated 10 propolis samples, Stoia et al. reported a much lower total phenolic content of 9.71 ± 0.80 mg GAE/g for methanol (95% *v/v*) extracts [13]. Similarly, low amounts of polyphenols, ranging between 24.46 and 62.39 g standard mixture/100 g propolis, was reported by Mihai et al. in 20 samples of Transilvanian propolis [14]. However, variations in the phenolic content are widely encountered in the literature and have a plurifactorial etiology: the solvent and technique used for the phenolic extraction, the storage and environmental conditions (temperature, season of collection, migratory or stationary apiaries, vegetation in the vicinity of the hives); the diversity in the chemical composition of propolis is an advantage that is responsible for its multiple biological effects [15]. Large variations in the phenolic content were reported among propolis samples harvested from temperate, tropical, and subtropical areas. In 15 Azerbaijan propolis samples (ethanol 95% extracts) the total phenolic content was between 10.94 to 79.23 mg GAE/g propolis, with an average value of 47.67 ± 5.14 mg GAE/g [16]. Two propolis samples (methanol extracts) harvested from two regions of Turkey were reported to contain 40.83 and 94.54 mg GAE/g propolis, respectively [17]. In 2 propolis methanol extracts from 2 regions of Portugal it was reported 151–329 mg GAE/g [18]. Abubaker et al. (2017) found 10.07 and 11.13 mg GAE/g, respectively in 2 propolis extracts (methanol) from Sudan [19]. In 5 Ethiopian propolis extracts (ethanol 70%), the total phenols varied from 365 ± 37 mg to 1022 ± 60 mg GAE/g [20]. In 14 propolis extracts (methanol) from Argentina, phenols were between 32.5 mg to 334.9 mg GAE/g propolis [21]. Korean propolis (20 samples, ethanol 80% extracts) were reported to contain between 48.5 mg and 238.9 mg GAE/g propolis [22]. As for Brazilian propolis, a recent study that analyzed 6 extracts (80% ethanol) reported the following data for the TPC: 249.28 ± 0.01 mg GAE/g (brown propolis), 374.10 ± 0.01 mg GAE/g (green propolis) and 481.49 ± 0.02 mg GAE/g (red propolis), respectively. Of note, the extracts obtained by supercritical extraction from the same propolis samples contained lower amounts of total polyphenols, namely 113.41 ± 0.01 mg GAE/g for brown propolis, 174.31 ± 0.02 mg GAE/g for green propolis, 171.33 ± 0.01 mg GAE/g for red propolis [23].At variance from these data, Andrade et al. reported lower values for the total phenolics in the 3 types of Brazilian propolis as follows: 55.74 ± 0.48 mg GAE/g for brown propolis, 90.55 ± 1.52 mg GAE/g for green propolis, and 91.32 ± 0.49 mg GAE/g for red propolis [24]. Jiang et al. recently reported the composition of a novel propolis type from North-Eastern China whose polyphenolic content varied between 215.6 ± 0.4 and 316.8 ± 1.2 mg GAE/g and was rich in *p*-coumaric acid [25].

As for the individual polyphenols, four compounds were found in higher concentraions in all samples, namely kaempferol, quercetin, rosmarinic acid, and resveratrol, respectively (Table 1 and Figure 2). Kaempferol and quercetin have also been identified in propolis from Serbia, Italy and Slovenia; these two flavonoids are considered the most abundant in poplar type propolis, *Populus sp. (P. alba, P. tremula, P. nigra)* the major type of propolis in temperate zones [4]. Coneac et al. (2014) reported variable concentrations for kaempferol in propolis from Timiș County (Timișoara) depending on the ethanol concentration used for extraction; from three concentrations (20%, 60%, 96% *v/v*), ethanol 60% extracted the highest amount of polyphenols. In their study, kaempferol varied between 1.33 and 3.54 mg/g and quercetin between 1.25–2.50 mg/g, respectively [26]. In our study, P6 sample was harvested from the same county (Timiș) and much higher concentrations forkaempferol (123.40–158.11 mg/g) and quercetin (2.05–26.57 mg/g) were found. Wang et al. reported anticancer, antioxidant (IC_50_ = 0.01372 mg/mL), and anti-inflammatory activities for kaempferol [27]. Epidemiological studies have shown an inverse association between cancer and kaempferol intake [28]. Similarly, quercetin is one of the mostly studied flavonoids as individual compound due to a plethora of therapeutic effects (reviewed in ref. [29]. However, Kocot et al. correctly highlighted the occurrence of synergistic effects in case of the in vivo administration of the complex mixture of propolis for therapeutic purposes [9].

The identification of resveratrol, as novel bioactive compound in Romanian propolis, is presented for the first time in this paper. Resveratrol, a powerful protective stilbene derivative, accounted for up to19.77% from all polyphenols in our samples. This is the most relevant finding of the present study and appears to be a characteristic of Western Romanian propolis since it was not reported in other types of Romanian propolis (reviewed in ref. [30]). As for international studies, there is a single paper published in 2004 that mentioned the presence of resveratrol in an ethanolic extract of Italian propolis [31]. Interestingly, in a comprehensive review that analyzed the chemical composition of propolis worldwide, Huang et al. mentioned the presence of prenylated stilbenes (5-farnesyl-3′-hydroxyresveratrol, 4-prenyldihydroresveratrol and3-prenylresveratrol) in propolis samples from Australia, Brazil, Greece, Indonesia, and Kenya; in these samples *Macaranga* was the plant source used by *Apis mellifera* [32].

Resveratrol is probably the most important polyphenol studied for the complex protective effects in ageing, cardiovascular pathology, malignancies and, more recently, the emerging field of cardio-oncology [33,34,35]. In this respect, the identification of resveratrol in the Western Romanian propolis samples is an important finding.

The phenolic acid *p*-coumaric acid was detected only in sample P2, albeit in minute concentration—yet this sample had the highest antioxidant activity. Rutin was present in 5 out of the 8 samples, ranging from 1.03 ± 0.73 μg/mL to 10.11 ± 3.22 μg/mL of extract. Epicatechin was identified in low quantity in a couple of samples, representing between 0.24% and 0.56% from individual polyphenols. Gallic acid, protocatechuic acid, caffeic acid and ferulic acid were not detected in the analyzed samples.

As regarding the PCA analysis of the 8 propolis samples, we report a clusterization as follows: P3, P5; P6, P8; P1, P7 and P2, P4. Three clusters seemed very similar regarding the polyphenolic profile, the percentage of similarity being greater than 90% (P1/P7; P3/P5; P6/P8). Two samples, P2 and P4 represented the outliers, as their similarity with the rest of samples being very low. Analyzing their polyphenolic content, sample P2 is remarked for the highest content of individual polyphenols (1176.61 ± 161.59 µg/mL), whereas sample P4 presented the highest total phenolic content (333.83 ± 13.79 mg GAE/g). Samples P1, P7 were grouped with P3 and P5 as they showed a similarity of 56.10%.

### 3.2. Antioxidant Activity Assays

The percentage of DPPH free radical inhibition along with IC_50_ were determined. The IC_50_ varied from 0.0700 mg/mL for the P2 sample to 0.9945 mg/mL for P6 sample. Interestingly, sample P2 had a lower IC_50_ value when compared to ascorbic acid (IC_50_ = 0.0757 mg/mL), indicating the strongest anti-oxidant capacity. Whether this effect can be recapitulated in vivo, in experimental conditions associated with oxidative stress warrants further investigation. Mărghitaș et al. reported IC_50_ values between 0.3 mg/mL and 5.6 mg/mL for 13 propolis ethanolic extracts originating from Transilvania [36]. Belfar et al. reported a stronger antioxidant activity for 4 methanolic extracts of Algerian propolis with IC_50_ varying from 0.007 to 0.066 mg/mL which was lower as compared to the value (0.184 mg/mL) for ascorbic acid used as control [37]. For 10 ethanolic extracts of Indian propolis, IC_50_ varied between 0.33348 mg/mL and 0.60088 mg/mL, while for ascorbic acid was 0.28492 mg/mL [38]. Wang et al. reported IC_50_ values from 0.043 to 0.269 mg/mL for 20 samples of Korean propolis [22]. Sun et al. analyzed the antioxidant activity of Chinese propolis and reported IC_50_ values for different propolis extracts varied between 0.633 mg/mL and 13.798 mg/mL [10]. Guzman-Gutierrez recently reported a strong DPPH scavenging activity (IC_50_= 16.55 ± 0.87 μg/mL) for Mexican propolis (ethyl-acetate extraction) [39].

Ahn et al. assessed the composition of several propolis samples harvested from 12 regions of China and concluded they were similar to poplar-type propolis. The authors used 3 techniques for the assessment of the antioxidant activity of the Chinese propolis: the inhibition of linoleic acid oxidation by means of beta-carotene bleaching, the DPPH radical-scavenging activity and thescavenging activity on 2,20-azinobis(3-ethylbenzothiazoline-6-sulfonic acid) (ABTS) radical cation. All but one sample displayed a high antioxidant activity that was associated with the presence of caffeic acid, caffeic acid phethyl ester and ferulic acid [40]. In a similar elegant study, Nagawa et al. assessed the antioxidant activity of ethanolic propolis extracts collected from 14 countries all over the world and reported a large variation in DPPH radical scavenging activity (from ~10% to ~90%), with the most potent samples originating from Australia, China, Hungary, and New Zealand [41]. Interestingly, water extracts of Brazilian propolis were also reported to exert antioxidant activity; the DPPH scavenging activity dose-dependently varied between 23.7% to 43.5% with ascorbic acid being used as positive control [42]. Of note, these studies used at least two techniques for the in vitro assessment of the antioxidant activity, one of them being the DPPH assay.

In a recent comprehensive study, Di Marco et al. used two antiradical assays (DPPH and FRAP) to assess the antioxidant activity of 460 Italia honeys and reported the highest antioxidant activity for the dark honeys [43].

We also thought to use a second technique, FOX assay for the assessment of the % of H_2_O_2_ inhibition for the samples reported to have the highest and lowest DPPH scavenging activity, respectively. Interestingly, while both concentrations act as H_2_O_2_ scavengers (similarly to catalase) a superior antioxidant effect was found for the low dose as compared to the high one. A hypothesis was formulated in the literature regarding the hormetic effect of phytochemicals [44] as classically described for several drugs. Whether this is the case for propolis it is not known. Of note, the hormetic effect was recently reported for resveratrol [45,46].

Moreover, bee products, including propolis also contain fatty acids [47]; in particular, the effect of short hydroxy fatty acids (C8-C12) and dicarboxylic acids was reported by some (but not all) papers to contribute to the anti-oxidant activity [9].

A limit of the present study is that we did not analyze other chemical components of propolis, in relation with the antioxidant activity. Indeed, it has been earlier suggested that flavonoids are responsible for the biological activities of European propolis [48]. At variance, the antioxidant activity of Brazilian propolis was mainly due to the phenolic constituents (and not to the flavonoid component) [24]. Other authors also mentioned that fact that the levels of the chemical components in propolis extracts does not always directly reflect their biological activity [49]. Nevertheless, there is an unmet need for the standardization of the phenolic profile assessment in terms of both total content and individual specific compounds since not only the former but also the latter might contribute to the antioxidant role [50].

## 4. Materials and Methods

### 4.1. Propolis Samples Collection

Propolis samples of *Apis mellifera* origin (abbreviated P1→P8, (Figure 13) were collected from the Western Romaniaduring 2015–2016: P1—Iteu, Bihor (BH) (47°21′7″N 22°25′7″E), 2015; P2—Bocşa, Caraș-Severin (CS) (45°22′29″N 21°42′38″E), 2015; P3—Șiria, Arad (AR) (46°16′2″N 21°38′18″E), 2015; P4—Ineu, Arad (46°26′N 21°50′E),2015; P5—Sărăuad, Satu Mare (SM) (47°28′45″N 22°37′32″E), 2015; P6—Folea, Timiș (TM) (45°29′54″N 21°18′0″E), 2015; P7—Iteu, Bihor (47°21′7″N 22°25′7″E), 2016; P8—Bocşa, Caraș-Severin (45°22′29″N 21°42′38″E), 2016 (Figure 12).

The appearance of the 8 propolis extracts in relation to their sites of collection is presented in Figure 13.

### 4.2. Preparation of the Extracts

Raw propolis was kept in freezer and the cooled samples were grinded prior to the extracts’ preparation. Propolis extracts were prepared using ethanol 60% (*v/v*) as solvent (SC Chimreactiv SRL, Bucharest, Romania) at a ratio of 1:20 (g/mL). Samples were stirred for 60 minat ambient temperature using a platform shaker (Heidolph PROMAX 1020) and then filtered through a filter paper. Subsequent dilutions were prepared for the experiments.

### 4.3. Assessment of Total Phenolic Content by Folin-Ciocâlteu Method

The total phenolic content was determined according to the Folin-Ciocâlteu method referred to in [51]. A volume of 0.5 mL of each extract (0.5 mg/mL) was treated with 1.25 mL Folin-Ciocâlteu reagent (Merck, Germany) diluted 1:10 (*v*/*v*) with distilled water. The samples were kept at room temperature for 5 min and further treated with 1 mL Na_2_CO_3_ 60 g/L (Reactivul București, Romania). After incubation at 50 °C for 30 min, the absorbance was measured at 760 nm using a UV-VIS spectrophotometer (Analytic Jena Specord 205). Calibration curve was obtained using gallic acid as standard (0–200 μg/mL) and the calibration equation was y = 0.0173x + 0.1224 (R^2^ = 0.9986), where x is the gallic acid concentration in μg/mL and y is the absorbance. Results were expressed as mg GAE/g propolis (mean ± SEM).

### 4.4. Assessment of Individual Polyphenols by Liquid Chromatography-Mass Spectrometry (LC-MS)

The separation and identification of polyphenols was performed by means of LC-MS (Shimadzu 2010 EV, Kyoto, Japan) with electrospray ionization according to a technique described in ref. [52] and adapted after ref. [53]. The chromatographic system comprises a LC unit with a UV-VIS spectrophotometer detector (SPD-10A), a degasser, an autosampler and solvent delivery pumps (LC-10AD) connected in-line with a MS-2010 mass spectrometer. The reversed-phase separation was performed on an EC 150/2 NUCLEODUR C18 Gravity SB 150 mm × 2.0 mm column, particle size 5 µm (Macherey-Nagel GmbH & Co. KG, Germany) operating at 20 °C at 0.2 mL/min flow rate. The compounds were separated with gradient elution of A (aqueous formic acid, pH = 3) and B (acetonitrile and formic acid, pH = 3). The gradient program was: 5% B (0.01–20 min), 5–40% B (20.01–50 min), 40-95% B (50–55 min), 95% B (55–60 min). The injection volume was 20 μL. Monitoring was performed at 280 and 320 nm and the detector was set at an acquisition range from 200 nm to 700 nm. The spectral acquisition rate was 1.25 scans/s (peak width: 0.2 min). Data acquisition, peak integration, and calibrations were performed with LC Solution software from Shimadzu. The calibration curves were performed in the range of 20–50 µg/mL. The measurements were performed in triplicate and the LC-MS analysis was conducted in the ESI positive mode (limit of detection 0.4–0.5 µg/mL, limit of cuantification 0.6–0.7 µg/mL). The results were expressed as mean value ±SEM of three parallel determinations for the 5 mg/mL extracts.

### 4.5. Assessment of the Antioxidant Capacity by DPPH (2,2-diphenyl-1-picrylhydrazyl) Assay

The DPPH assay represents a classic method frequently used to assess the antioxidant capacity of plant extracts that was adapted from ref. [54]. Moacă et al. standardized the technique originally described in refs. [55,56] using the DPPH reagent (Sigma-Aldrich, Germany, batch no.: # STBF5255V) and a UV-Line 9400 spectrophotometer (SI Analytics) at the Faculty of Pharmacy of the University of Medicine and Pharmacy of Timisoara, RO and performed the initial sample analysis (data not shown). In the present study, a volume of 0.5 mL of each extract was added to 2 mL ethanol 60% (*v/v*) and to 0.5 mL DPPH (Calbiochem^®^, EMD Millipore Corp., Billerica, MA, USA, batch: D00174004) 1 mM ethanol solution. The reaction was automatically monitored for 1200 s at 517 nm on a UV-VIS spectrophotometer (Analytic Jena Specord 205). The absorbance was continuously measured from 5 to 5 s. Ascorbic acid 0.13 mg/mL in ethanol 60% (*v/v*) was used as positive control. Ascorbic acid was purchased from Lach-Ner Company (Czech Republic). Radical scavenging activity (RSA) was calculated with the formula: RSA (%) = 100 − (A_517 (sample)_/A_517 (DPPH)_) × 100), where RSA = radical scavenging activity of extract (%), A_517 (sample)_ = sample absorbance measured at 517 nm at time *t*, A_517 (DPPH)_ = the absorbance of DPPH solution measured at 517 nm at time *t.* The antioxidant capacity of the extracts was expressed as the IC_50_ value and compared to the one of ascorbic acid.

### 4.6. Assessment of the Antioxidant Capacity by FOX Assay

The FOX assay was performed according to the method described in ref. [57]. The principle of the assay is as follows: under acidic conditions peroxides will convert Fe^2+^ to Fe^3+^ ions which will then form a colored adduct with xylenol orange (XO) measurable at 560 nm. The reaction can be described as: Fe^2+^ + R-OOH → Fe^3+^ + RO·+ OH^−^ and Fe^3+^ + XO →Fe^3+^-XO (blue colored adduct), where: XO = xylenol orange and R = H or a lipidic group.A standard solution of 100 μM hydrogen peroxide (H_2_O_2_) was firstly prepared, followed by the preparation of the working color reagent (by mixing 100 volumes of aqueous peroxide color reagent with 1 volume of ferrous ammonium sulfate reagent according to the manufacturer instructions (PeroxiDetect kit, Sigma Aldrich). 

Polyethylene-glycol (PEG)-catalase (100 U/mL, Sigma Aldrich), a classic H_2_O_2_ scavenger, was used as positive control. A volume 100 μL of propolis sample plus 100 μL of standard hydrogen peroxide solution were mixed with 2 mL of working color reagent and incubated at room temperature (22–25 °C) for ~30 min. Samples (in duplicate) were read at 560 nm (spectrophotometer Jenway 6100). The results were expressed as % of hydrogen peroxide inhibition.

### 4.7. Statistical Analysis

Results were expressed as mean ±standard error of the mean (SEM) as descriptive statistics. For the comparison of numerical values’ distribution across the samples, one-way ANOVA test followed by Bonferroni-adjusted multiple-comparisons between the pairs of samples and *t* test were used when appropriate. Inter-sample correlation and was conducted to investigate the underlying similarity of polyphenolic profiles. Kaiser-Meyer-Olkin (KMO) measure of sampling adequacy and Bartlett’s test of sphericity were applied to verify the data suitability for PCA. The scree plots and the eigenvalues over 1 were considered as criteria for deciding the appropriate number of components to be extracted. Based on PCA extracted components, a hierarchical clustering was applied, using Ward’s minimum variance method and squared Euclidian distance. The Pearson linear correlation coefficients were supplementary determined and analyzed for certain variables describing the antioxidant activity. All reported probability values were two-tailed and a 0.05 level of significance was considered, while marking the highly significant values (i.e., *p* < 0.01 and *p* < 0.001) as well. GraphPad Prism 7 and Minitab 18 were employed for data analysis.

## 5. Conclusions

The present study firstly reports the presence of resveratrol as a novel and potent bioactive molecule in the composition of Western Romanian propolis; its contribution to the beneficial biological properties of individual propolis samples warrants further investigation. The polyphenolic profile of propolis samples from Western Romania was characterized and, based on PCA analysis, clusters with a high level of similarity were identified.

## Figures and Tables

**Figure 1 molecules-24-03368-f001:**
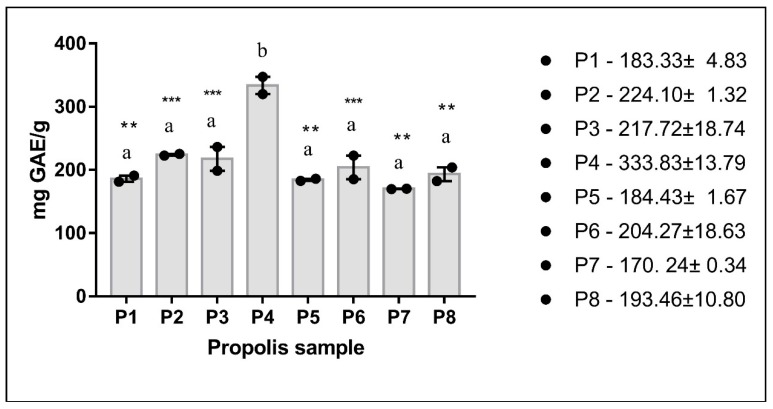
Total phenolic content of propolis samples P1-P8. Distinct letters on graph indicate significant differences among samples (** *p* < 0.01, *** *p* < 0.001, Bonferroni). (Data are expressed as mean ± SEM, b indicates significant difference in TPC from samples marked with a).

**Figure 2 molecules-24-03368-f002:**
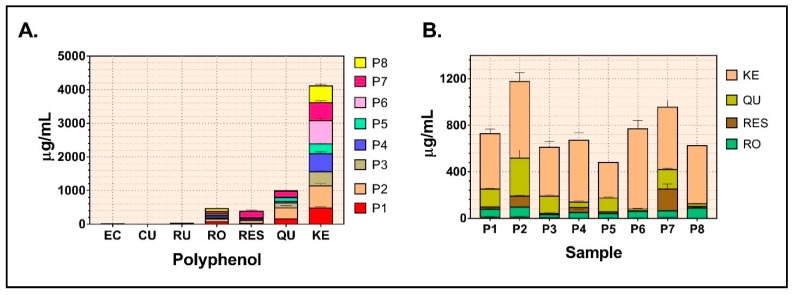
The individual polyphenols detected in propolis samples P1-P8 (**A**) and the main 4 phenolic compounds identified (**B**). (Data are expressed as mean ± SEM). (EC epicathechin, CU-*p*-coumaric acid, RU-rutin, RO-rosmarinic acid, RES-resveratrol, QU-quercetin, KE-kaempferol).

**Figure 3 molecules-24-03368-f003:**
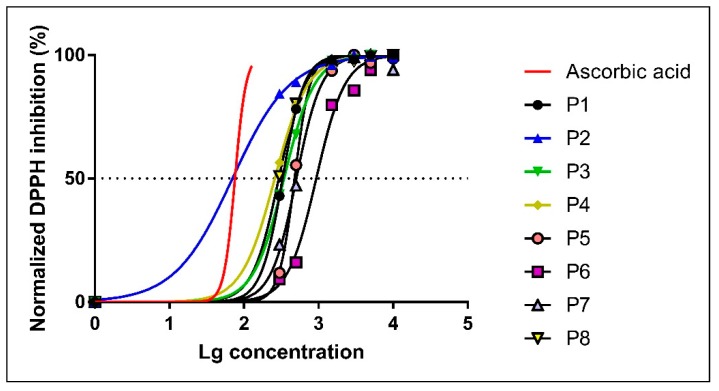
Dose-response curves for DPPH radical scavenging activity of the propolis samples.

**Figure 4 molecules-24-03368-f004:**
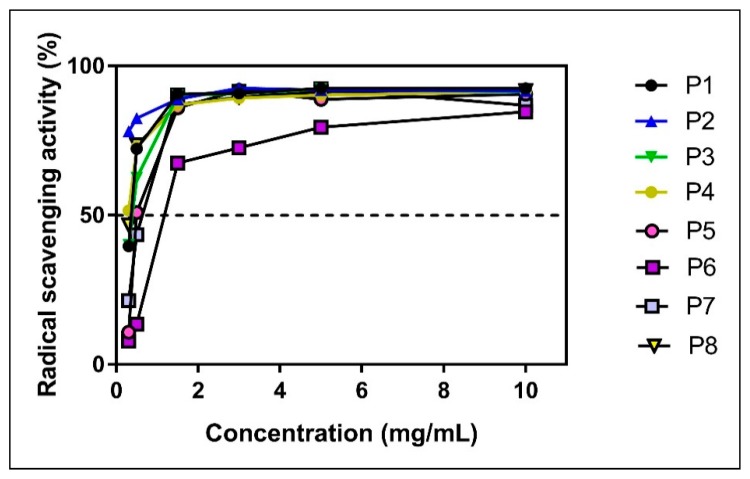
Concentration-dependency of radical scavenging activity of propolis samples (DPPH assay).

**Figure 5 molecules-24-03368-f005:**
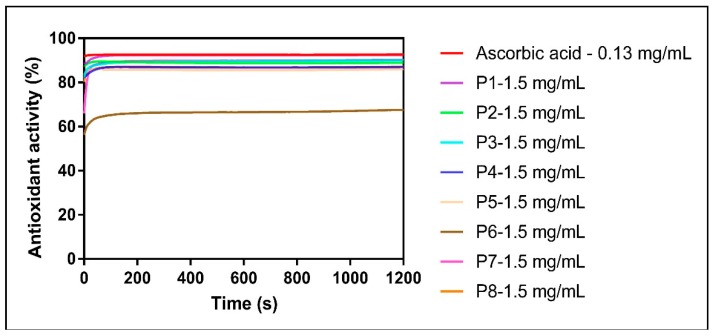
Time-dependency of radical scavenging activity of propolis (1.5 mg/mL) as compared with the standard antioxidant (DPPH assay).

**Figure 6 molecules-24-03368-f006:**
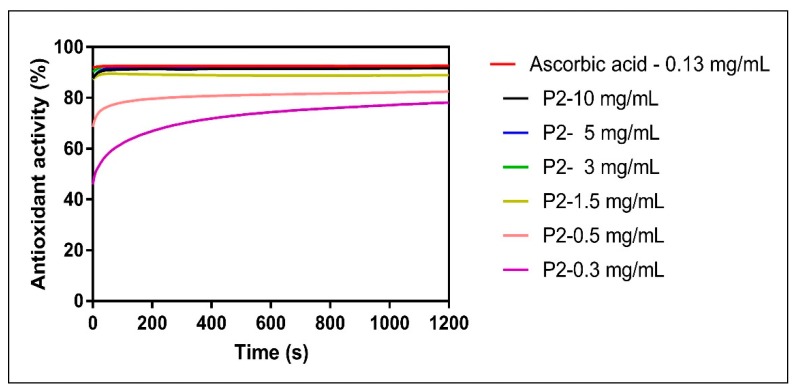
The time-dependency of radical scavenging activity for P2 sample vs. ascorbic acid.

**Figure 7 molecules-24-03368-f007:**
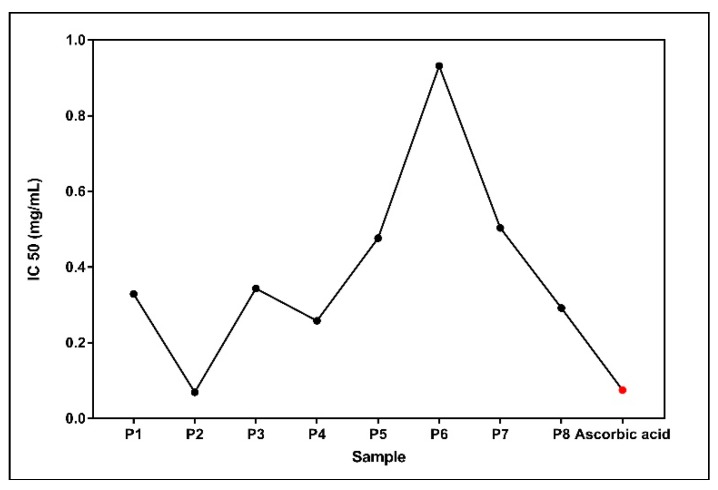
IC_50_ variation among P1-P8 samples.

**Figure 8 molecules-24-03368-f008:**
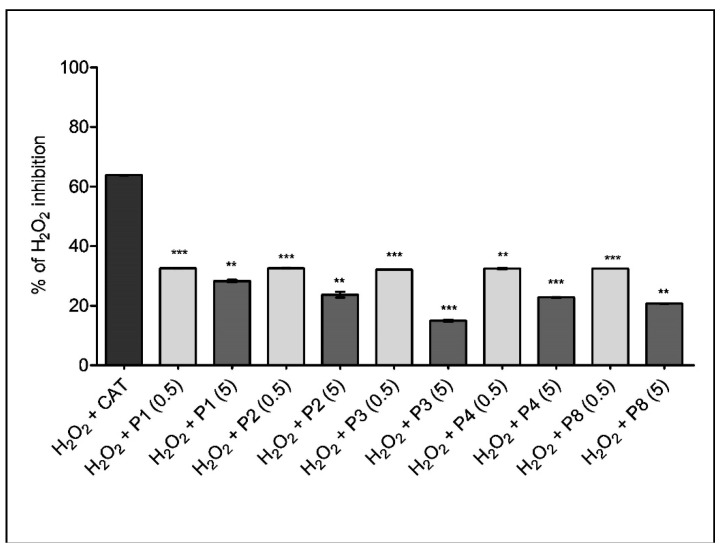
FOX assay for samples P1, P2, P3, P4, and P8 applied in 2 concentrations, 5 mg/mL and 0.5 mg/mL (** *p* < 0.01, *** *p* < 0.001 vs. CAT).

**Figure 9 molecules-24-03368-f009:**
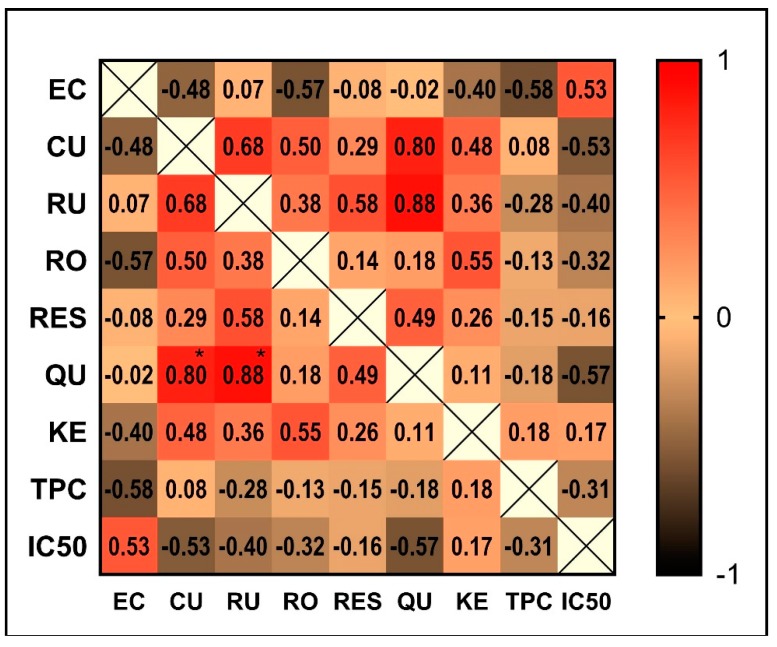
The inter-sample correlation matrix (* *p* < 0.05). (EC-epicatechin, CU-*p*-coumaric acid, RU-rutin, RO-rosmarinic acid, RES-resveratrol, QU-quercetin, KE-kaempferol, TPC-total phenolic content).

**Figure 10 molecules-24-03368-f010:**
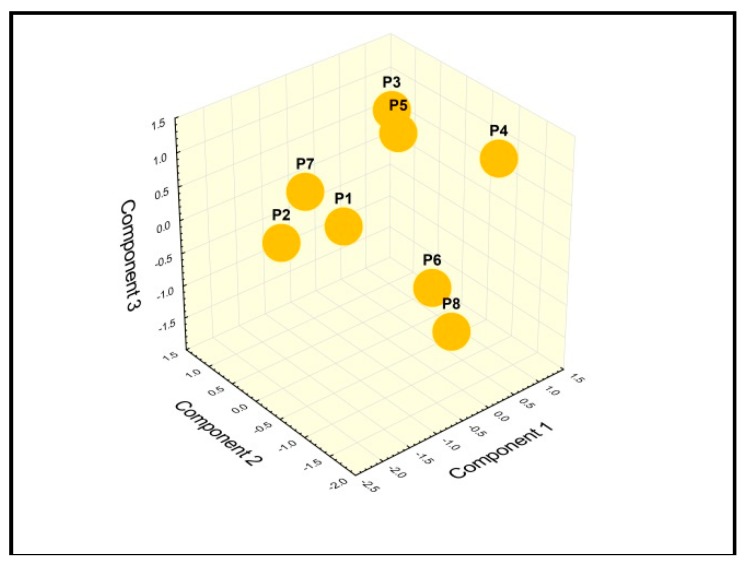
Cluster analysis—projection of the samples on a plane spanned by the three principal components.

**Figure 11 molecules-24-03368-f011:**
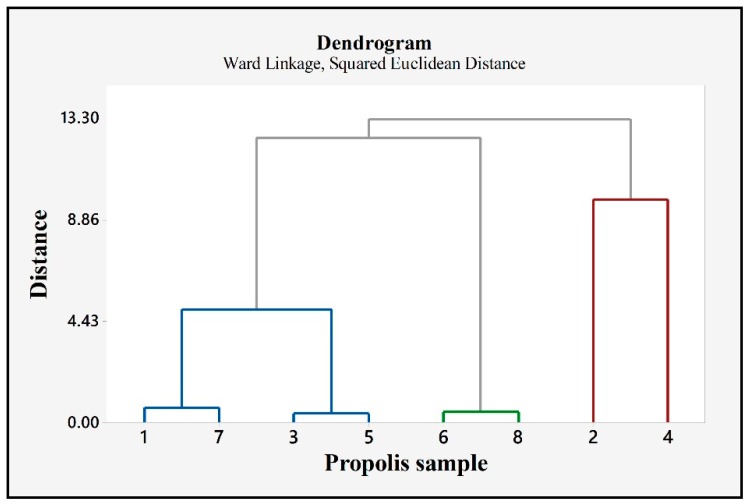
Hierarchical cluster analysis of the samples.

**Figure 12 molecules-24-03368-f012:**
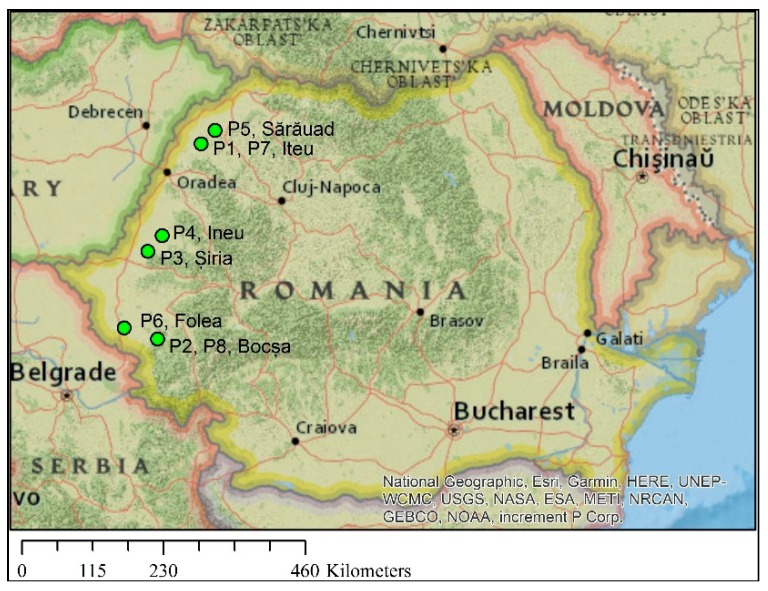
The geographic origin of the propolis samples (green dots).

**Figure 13 molecules-24-03368-f013:**
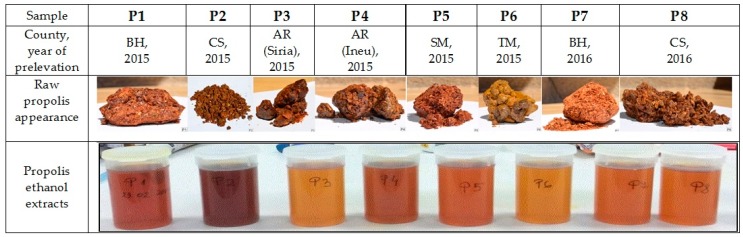
Collection sites and macroscopic appearance for propolis samples P1–P8. (BH—Bihor County, CS—Caraș-Severin County, AR—Arad County, SM—Satu-Mare County, TM—Timiș County).

**Table 1 molecules-24-03368-t001:** The distribution of the polyphenolic compounds in propolis extracts (LC-MS assay).

	**RT**	**m/z**		**P1**				**P2**				**P3**				**P4**		
				**μg/mL**		**%**		**μg/mL**		**%**		**μg/mL**		**%**		**μg/mL**		**%**
***Gallic acid***	5.175	169		nd		0.00		nd		0.00		nd		0.00		nd		0.00
***Protocatechuic acid***	11.112	153		nd		0.00		nd		0.00		nd		0.00		nd		0.00
***Caffeic acid***	22.341	179		nd		0.00		nd		0.00		nd		0.00		nd		0.00
***Epicatechin***	22.205	289		3.16 ± 0.97		0.43		nd		0.00		2.56 ± 1.82		0.42		nd		0.00
***p-Coumaric acid***	24.316	163		nd		0.00		0.05 ± 0.03		0.00		nd		0.00		nd		0.00
***Ferulic acid***	24.753	193		nd		0.00		nd		0.00		nd		0.00		nd		0.00
***Rutin***	25.910	609		7.15 ± 3.03		0.98		10.11 ± 3.22		0.86		2.94 ± 1.82		0.48		nd		0.00
***Rosmarinic acid***	29.289	359		67.81 ± 5.38		9.33		84.81 ± 6.03		7.20		26.41 ± 1.98		4.34		49.11 ± 27.85		7.34
***Resveratrol***	29.442	227		18.66 ± 2.79		2.57		94.34 ± 8.44		8.01		10.04 ± 4.26		1.65		43.69 ± 26.67		6.53
***Quercetin***	31.650	301		154.46 ± 7.16		21.27		328.35 ± 68.46		27.91		146.44 ± 12.18		24.06		44.65 ± 12.25		6.68
***Kaempferol***	34.535	285		475.07 ± 40.67		65.41		658.94 ± 75.40		56.00		420.19 ± 51.18		69.04		531.38 ± 64.14		79.45
**Total**	-	-		726.31 ± 60.00		100		1176.61 ± 161.59		100		608.58 ± 73.24		100		668.83 ± 130.91		100
			**P5**				**P6**				**P7**				**P8**			
			**μg/mL**		**%**		**μg/mL**		**%**		**μg/mL**		**%**		**μg/mL**		**%**	
***Gallic acid (GA)***			nd		0.00		nd		0.00		nd		0.00		nd		0.00	
***Protocatechuic acid (PA)***			nd		0.00		nd		0.00		nd		0.00		nd		0.00	
***Caffeic acid (CA)***			nd		0.00		nd		0.00		nd		0.00		nd		0.00	
***Epicatechin (EC)***			2.69 ± 1.90		0.56		2.31 ± 1.70		0.30		2.38 ± 1.69		0.25		nd		0.00	
***p-Coumaric acid (CU)***			nd		0.00		nd		0.00		nd		0.00		nd		0.00	
***Ferulic acid (FE)***			nd		0.00		nd		0.00		nd		0.00		nd		0.00	
***Rutin (RU)***			nd		0.00		1.03 ± 0.73		0.13		6.64 ± 1.55		0.70		nd		0.00	
***Rosmarinic acid (RO)***			36.82 ± 3.59		7.69		57.18 ± 7.16		7.46		54.57 ± 4.46		5.72		87.53 ± 6.11		14.04	
***Resveratrol (RES)***			15.91 ± 2.42		3.32		4.90 ± 0.57		0.64		188.50 ± 42.52		19.77		12.69 ± 0.83		2.03	
***Quercetin (QU)***			118.49 ± 8.11		24.73		15.07 ± 4.80		1.97		167.07 ± 11.67		17.52		22.61 ± 10.28		3.62	
***Kaempferol (KE)***			305.21 ± 23.11		63.70		686.11 ± 75.11		85.50		534.52 ± 57.35		56.05		500.78 ± 47.51		80.30	
**Total**			479.12 ± 39.13		100		766.59 ± 90.07		100		953.69 ± 119.25		100		623.61 ± 64.75		100	

Legend: RT = retention time, m/z = mass to charge ratio, nd = not detected. Data are expressed as mean ± SEM of three independent measurements.

**Table 2 molecules-24-03368-t002:** The DPPH radical scavenging activity (% inhibition) of propolis ethanolic extracts vs. ascorbic acid.

Propolis									Ascorbic Acid	
Concentration (mg/mL)	P1	P2	P3	P4	P5	P6	P7	P8	Concentration (mg/mL)	% Inhibition
	% Inhibition									
10	92.57	91.75	91.44	91.48	90.62	84.71	86.86	91.78	0.13	92.68
5	92.50	92.05	92.26	90.30	88.90	79.57	92.34	91.37	0.11	91.71
3	90.84	92.63	90.80	89.33	91.85	72.63	91.56	89.63	0.105	89.58
1.5	90.66	88.97	90.15	87.02	86.01	67.58	90.26	87.12	0.10	72.22
0.5	72.31	82.52	62.57	72.88	50.94	13.58	43.57	73.66	0.08	43.28
0.3	39.74	78.16	40.06	51.65	10.85	7.89	21.45	46.79	0.06	27.68

**Table 3 molecules-24-03368-t003:** The IC_50_ value of propolis samples vs. ascorbic acid (best-fit values *).

Propolis	P1	P2	P3	P4	P5	P6	P7	P8	Ascorbic Acid
**IC_50_ ± SEM (mg/mL)**	0.3292 ± 0.0035	0.0700 ± 0.0132	0.3439 ± 0.0059	0.2586 ± 0.0089	0.4770 ± 0.0127	0.9320 ± 0.0760	0.5039± 0.0234	0.2925 ± 0.0092	0.0757 ± 0.0037
**R^2^**	0.9993	0.9993	0.9989	0.9962	0.9899	0.9945	0.9979	0.9993	0.9531
**Hill Slope**	3.073	0.9313	2.111	2.128	4.299	2.649	2.679	2.663	5.869

* log inhibitor vs. normalized response, variable slope.

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
