# Peer review of "Identification of Resveratrol as Bioactive Compound of Propolis from Western Romania and Characterization of Phenolic Profile and Antioxidant Activity of Ethanolic Extracts"

_molecules, 2019, doi:10.3390/molecules24183368_

Round 1
Reviewer 1 Report
The paper entitled “Characterization of the Phenolic Profile and Antioxidant Activity of Ethanolic Propolis From Western Romania: Identification of Resveratrol as Bioactive Compound” is very interesting because it chemically characterized several Romanian propolis and their relative antioxidant capacity. In particular, the resveratrol was detected and identified in them. I have some suggestions for the authors to improve their paper.
1 Authors report that “Data are expressed as mean ± SE of two independent measurements” but how many measurements were performed in total? What was n value? They should replicate at least three times each experiment.
2 The caption of figure 2 should be described better. Each acronym reported in the image should be described.
3 The identification of resveratrol in propolis has been already reported by Volpi, N. (2004, Separation of flavonoids and phenolic acids from propolis by capillary zone electrophoresis. Electrophoresis, 25, 12, 1872-1878). Therefore, what is the innovative aspect of the present work? Why did the authors associate the bioactivity of the propolis samples to resveratrol if, in some cases, they showed just a little amount of this specific plant secondary metabolite?
4 A representative chromatogram should be reported for each sample, as supplemental data.
5 As in some case authors were not able to detect some secondary metabolites by LC-MS, what is the limit of detection and the sensibility of the instrument?
6 The data about the antioxidant properties of the propolis studied in the present work should be compared with the others reported in literature and also with those relative to honey and commented. I suggest to cite the following papers: Food chemistry, 2004, 84(3), 329-339; Food chemistry, 2003, 80(1), 29-33; Journal of food science and technology, 2018, 55(10), 4042-4050; Food Chemistry, 2007 101(4), 1383-1392.
Reviewer 2 Report
The manuscript brings the antioxidant activity and the phenolic profile of eight propolis from Western Romania. Also, the paper highlighted the presence of the stilbene resveratrol in these samples, which the authors attribute to a specific characteristic of the propolis from this region of Romania. However, there are some issues that must be solved before considering the manuscript for publication in Molecules.
Title:
Suggestion: Identification of resveratrol as bioactive compound of propolis from Western Romania.
The modification of the title aims to highlight the presence of resveratrol in these propolis. Antioxidant activity and phenolic profile of propolis are mandatory, being trivial information. The suggestion is that the authors should focus on resveratrol to enforce on innovative about these propolis.
Abstract
Line 26: results of total phenolic are expressed as g of propolis or extract?
Line 29: Why did the authors choose to use ascorbic acid as standard?
Line 33: Why did not the author use all samples for FOX assay?
Introduction
Lines 62-66:
The authors should include the identification of resveratrol (highlight in polyphenols profile) in the objectives.
Results
Figure 2: include name of the compounds in the legend or in the title of the figure. Also, the title should be citing the graphic “B”
Table 1: Please, cite the technique used for identification in the title of the table.
The analysis of LC-MS was conducted in positive, negative or both?
Please, review the deviation standards of the results of phenolic compounds. Most of them are unacceptable and nullify these results.
Table 2: concentration of samples refers to mg of extract or propolis?
Table 3: What “best-fit values” means?
Line 147: the samples are not comparable to standard. This affirmation is not true. The concentrations used to obtain similar values are different.
Figure 3: Please cite the assay used to produce these results (showed in the figure)
Line 161: Again, the results are not comparable. Review the sentences along the manuscript.
Figure 4: Were results of the graphic measured by DPPH? Please, include this information
Lines 211-215: It is suggested that the authors analyze the resveratrol (standard) to allow better discussion of the results, including advances on the knowledge about the contribution of the resveratrol to the antioxidant activity of these propolis.
Lines 222-223: It is not clear the relation between H2O2 and DPPH.
Line 244: please, remove “comparable”
PCA analysis: Why did the authors not analyze the correlation between phenolic compounds and antioxidant activity?
Lines 247-258 and Figure 8: please, provide the meaning of the abbreviations.
Lines 268-269: “P6 and P8 … with a similarity of 72.98%” is repeated in the sentence.
Discussion
Line 288: remove “from”.
Lines 283-284: The statement contradict the affirmation made at lines 211-215. Please check it.
Line 324: p-coumaric (p is italic)
Line 351: Apis mellifera in italic
Page 14: please, include discussion about PCA.
Materials and methods
Line 401: include animal origin of the propolis (Apis mellifera?)
Line 426: did propolis were grinded cooled? Please explain it.
Still about extraction: What was the temperature? Was beeswax separated from extract?
Line 433: “phenolic”
Line 434: The concentration of 0.5 mg/ml means that the extract was diluted 100 times?
Line 443: please include the mode which the samples were analyzed in LC-MS (positive, negative)
Line 461: why did the authors chose DPPH method? DPPH is a method in disuse because it does not represent most of the reactions that occur naturally once it uses a synthetic free radical.
Still about DPPH: why did the authors use different spectrophotometers (lines 465 and 470). The measures each 5 s indicates that the readings were automatized. Is it true?
Conclusion
There is no conclusion about the PCA analyzes. Include it.
The authors should analyze the standard of resveratrol to allow more consistency for the discussion and conclusion of this research.
Reviewer 3 Report
The present study assesses antioxidant activity with a DPPH inhibition assay followed by the FOX assay. The IC50 variance in the sample P6 from the DPPH assay suggest something unique about the chemical composition of that sample relative to the other samples. I wonder why the authors did not attempt to study P6 with the FOX assay considering this finding. This would bring greater significance to the paper.
Also, as the authors mention, a full chemical analysis has yet to be conducted for the assays. The flavonoid content will allow the reader to better understand and relate the chemical composition of the organic extract to the antioxidant activity of the samples.
The manuscript is appropriate for Molecules in its present form.
Round 2
Reviewer 2 Report
The authors made the required modifications and the manuscript now is able to be published.